# Detection of Elevated Level of Tetrahydrobiopterin in Serum Samples of ME/CFS Patients with Orthostatic Intolerance: A Pilot Study

**DOI:** 10.3390/ijms24108713

**Published:** 2023-05-13

**Authors:** Carl Gunnar Gottschalk, Ryan Whelan, Daniel Peterson, Avik Roy

**Affiliations:** 1Simmaron Research Institute, 948 Incline Way, Incline Village, NV 89451, USA; 2Simmaron Research and Development Laboratory, Chemistry Building, University of Wisconsin-Milwaukee, 3210 N Cramer Street, Suite # 214, Milwaukee, WI 53211, USA; 3Sierra Internal Medicine, 920 Incline Way, Incline Village, NV 89451, USA

**Keywords:** ME/CFS, tetrahydrobiopterin, orthostatic intolerance, small fiber polyneuropathy, endothelial NOS, reactive oxygen species

## Abstract

Myalgic encephalomyelitis or chronic fatigue syndrome (ME/CFS) is a multisystem chronic illness characterized by severe muscle fatigue, pain, dizziness, and brain fog. Many patients with ME/CFS experience orthostatic intolerance (OI), which is characterized by frequent dizziness, light-headedness, and feeling faint while maintaining an upright posture. Despite intense investigation, the molecular mechanism of this debilitating condition is still unknown. OI is often manifested by cardiovascular alterations, such as reduced cerebral blood flow, reduced blood pressure, and diminished heart rate. The bioavailability of tetrahydrobiopterin (BH4), an essential cofactor of endothelial nitric oxide synthase (eNOS) enzyme, is tightly coupled with cardiovascular health and circulation. To explore the role of BH4 in ME/CFS, serum samples of CFS patients (*n* = 32), CFS patients with OI only (*n* = 10; CFS + OI), and CFS patients with both OI and small fiber polyneuropathy (*n* = 12; CFS + OI + SFN) were subjected to BH4 ELISA. Interestingly, our results revealed that the BH4 expression is significantly high in CFS, CFS + OI, and CFS + OI + SFN patients compared to age-/gender-matched controls. Finally, a ROS production assay in cultured microglial cells followed by Pearson correlation statistics indicated that the elevated BH4 in serum samples of CFS + OI patients might be associated with the oxidative stress response. These findings suggest that the regulation of BH4 metabolism could be a promising target for understanding the molecular mechanism of CFS and CFS with OI.

## 1. Introduction

Myalgic Encephalomyelitis/Chronic Fatigue Syndrome (ME/CFS) is a chronic multisystem disease which is characterized by debilitating muscle fatigue, pain, dizziness, and brain fog [1,2]. A subset of ME/CFS patients display severe orthostatic hypotension [3] characterized by a significant drop in blood pressure causing dizziness and fainting while standing in an upright condition [4]. This condition is often termed orthostatic intolerance (OI). A subset of OI patients also meet the diagnostic criteria for small fiber polyneuropathy [5,6] that is diagnosed using Congo red staining for amyloidosis in skin biopsy samples [7]. Despite intense investigation, the molecular mechanism of OI is not known.

BH4 is an important cofactor of amino acid metabolism [8]. Amino acids, such as phenylalanine, tyrosine, and tryptophan, are converted into important cellular intermediates by the action of hydroxylase enzymes. BH4 serves as an essential cofactor of all these hydroxylase enzymes [9,10,11]. Accordingly, the bioavailability of BH4 is critical for the synthesis of neurotransmitters, such as dopamine and serotonin [10]. Apart from these hydroxylase enzymes, BH4 also regulates the function of nitric oxide synthase enzymes and controls the production of nitric oxide (NO) in endothelium [12]. Endothelial production of NO enhances the vasodilative response, lowers blood pressure, and helps muscle relaxation [13,14]. On the other hand, in the presence of oxidative stress, BH4 uncouples from eNOS-induced NO production [15] and directly augments the mitochondrial stress response in cardiac tissue [16] and skeletal muscle [17]. Therefore, the regulation of BH4 metabolism is critical for cardiovascular function and circulatory health. Since OI patients are reported to display cardiovascular abnormalities, we are interested in studying if OI patients have altered BH4 metabolism. Until now, there has been no study that explores the role of BH4 metabolism in CFS. Moreover, our study also explores if CFS patients with OI have altered BH4 biosynthesis.

Serum samples of 66 subjects were included in the study with *n* = 32 CFS and *n* = 34 controls. A double-blinded quantitative ELISA was performed to compare BH4 levels between healthy controls and CFS patients. Interestingly, we observed that there was a strong upregulation of BH4 in CFS patients compared to healthy controls. While comparing the level of BH4 between CFS patients with OI (*n* = 10) and age-/gender-matched controls (*n* = 10), we also observed that CFS patients with OI had a significantly higher level of BH4 in serum. Collectively, our current manuscript highlights that the elevation of serum BH4 in CFS patients could play a critical role in the pathogenesis of OI.

## 2. Results

*Subjects included in the study:* To quantify BH4 level in serum samples, a total of 66 human subjects were included in the study. Table 1 summarizes the age, gender, ethnicity, and pre-existing conditions of these subjects. Four subjects from the control group had cancer-related complications and therefore were excluded from the final analysis. The minimum, median, and maximum age of control group are 18, 64.5, and 82 yrs, respectively. For CFS group, these values are 26, 59, and 76 yrs. A frequency distribution plot of age in both groups exhibited a similar skewness and a subsequent Bland–Altman analysis test revealed that the average of the differences was close to zero suggesting that there is no bias with respect to age. The control group consisted of 44% males and 56% females, whereas the CFS group consisted of 41% males and 59% females. All patients were Caucasians and lived in the geographical region of Northern Nevada and California.

The ME/CFS patient population was enrolled based on the Canadian 2003 and Fakuda criteria for ME/CFS. They also met the diagnostic criteria for SCID (ME/CFS’s new name established by the Institute of Medicine). Controls were age/gender matched to CFS subjects who were also invited along with patients in Dr. Peterson’s Internal Medicine Clinic.

CFS + OI and CFS + OI_SFN subsets were established via diagnostic testing records (medical chart review-complete) and clinical correlation. For example, people with OI presented to the office with a modified positive NASA lean test (showing the OI) and all received IV volume expansion therapy (saline). SFN patients had biopsy samples. For the included CFS + OI + SFN subjects, the Mass General Hospital report was recorded, in which Congo red staining to count neurite density was performed in the lateral ankle muscle tissue.

*BH4 level is high in the serum samples of ME/CFS patients:* In total, 32 CFS and 34 control subjects were included in the study (Table 1) to measure serum BH4 levels by competitive ELISA. The standard curve of BH4 ELISA was a negative slope exponential curve that was derived from six increasing concentrations of BH4 standards (5, 10, 25, 50, 100, and 150 ng/mL). The slope and intercept of the standard curve were used to measure absolute levels of BH4 from the OD value. A serial dilution of serum samples followed by BH4 ELISA assay revealed that a 1:2 dilution of serum provided the most contrasted results compared to blank or background. Therefore, all serum samples were subjected to 1:2 dilution. Interestingly, a double-blinded ELISA analysis followed by a non-parametric Mann–Whitney *U* test [* *p* < 0.05 (=0.033); U = 304.5] revealed that the levels of BH4 were significantly higher in *n* = 32 CFS patients compared to *n* = 28 control subjects (Figure 1A). Four subjects (*n* = 4) in the control group were excluded because of their pre-existing cancers. Since subjects were included randomly irrespective of their age and sex, we next wanted to analyze the correlation of BH4 level with age and gender of the subjects. Accordingly, a non-parametric Spearman correlation analysis (Figure 1B) revealed that there was no correlation between BH4 level and the age of subjects’ (Spearman r = −0.1657; *p* = 0.1835). Next, we compared the level of BH4 between male and female subjects (Figure 1C). To assess the significance between male and female (effector#1 is gender), as well as between control and CFS (effector#2 is health condition), a two-way ANOVA analysis was performed. Accordingly, male and female subjects did not show any significant difference in BH4 levels (*p* > 0.05; =0.3496). However, there was a significant difference of BH4 between control and CFS subjects (*p* < 0.05; =0.0375) irrespective of the sex difference.

Since the current cohort is strictly restricted under the same geographical area, ethnicity, and from a single physician’s practice, a pair-wise comparison analysis of BH4 between cases and controls matching age and gender would be more conclusive. A total of 15 pairs were compared (Figure 1D). A non-parametric Wilcoxon matched-pairs signed rank test revealed that CFS patients had significantly higher BH4 (** *p* <0.01; =0.0015) than age- and gender-matched controls. Collectively, our results suggest that the BH4 level is significantly higher in CFS patients compared to healthy control subjects, irrespective of age and gender differences.

*CFS patients with OI have upregulated levels of BH4 in serum:* Next, we wanted to analyze BH4 levels between control (*n* = 10) and CFS patients with OI (*n* = 10). An unpaired *t*-test revealed that CFS + OI patients had significantly higher BH4 [t_1,18_ = 2.499; * *p* < 0.05 (=0.0223)] levels compared to control subjects (Figure 2A). Table 2 displays that all controls were carefully matched with CFS + OI subjects in terms of age and gender. Moreover, a Kolmogorov–Smirnov normality distribution analysis followed by a Q-Q (quantile-quantile) plot (Figure 2B) indicated that all datapoints were normally distributed [*p* >alpha (0.05)]. Since all CFS and control samples were matched with age and gender, next a pairwise comparison analysis was performed. The subsequent paired *t*-test (t_1,9_ = 3.834; ** *p* < 0.01 = 0.004) indicated that CFS + OI serum samples had significantly higher BH4 compared to control serum samples (Figure 2C). A cumulative column statistics test (Figure 2D) further revealed that 7 out of 10 CFS + OI patients had significantly higher BH4 levels than median (=99.63 ng/mL). On the other hand, only 2 out of 10 control subjects had higher BH4 than median value. Collectively, these data suggest that CFS + OI patients had significantly higher BH4 compared to control subjects.

A subset of CFS patients (*n* = 12) have both OI and SFN. A comparison of BH4 levels (Figure 3A) between healthy control and CFS + OI + SFN subjects revealed that CFS + OI + SFN patients had a significantly higher level of BH4 [t_1,22_ = 2.371; * *p* <0.05 (=0.0269)]. Table 3 shows that the cohort of 24 serum samples (*n* = 12 Control + *n* = 12 CFS + OI + SFN) was carefully selected based on the age and gender. Subsequently, a Q-Q plot demonstrated all datapoints from both the groups were normally distributed (Figure 3B). Since all subjects were matched in terms of the age and gender, a paired analysis of control and CFS + OI + SFN subjects were performed. Accordingly, significant difference in BH4 expression (* *p* < 0.05) was observed between these two groups (Figure 3C). A cumulative statistical analysis (Figure 3D) further indicated that *n =* 6 CFS + OI + SFN patients and *n* = 3 controls have higher BH4 levels than median (=81.83 ng/mL) suggesting that the BH4 level is strongly elevated in serum samples of CFS + OI + SFN subjects.

While comparing BH4 levels between CFS alone and CFS + OI patients, we observed that serum samples of CFS + OI patients (*n* = 14) had higher levels of BH4 (Figure 4A) compared to these of control subjects (*n* = 10). The Q-Q plot analysis displays that the distribution of data is normal in both groups (Figure 4B), suggesting the parametric test can be adopted to compare means. Accordingly, an unpaired *t*-test to analyze the significance of the mean between groups revealed a strong difference [t_1,22_ = 2.592; * *p* < 0.05 (=0.0166)]. Column statistics of the selected cohort (*n* = 22) indicated that 9 out of 14 CFS + OI patients had higher serum BH4 levels compared to a median of 119.7 ng/mL (Figure 4C). Taken together, our results suggest that CFS patients with OI have elevated levels of BH4 in serum. Collectively, our current manuscript identifies BH4 as a potential blood-borne biomarker for CFS + OI and CFS + OI + SFN patients.

*Elevated BH4 levels might be associated with increased oxidative stress*: To understand the functional importance of elevated BH4 in serum, we explored the ROS inducing ability of serum samples of CFS patients with OI (*n* = 10). DCFDA-infused microglial cells were treated with 1:2 diluted serum samples from 10 CFS + OI and 10 age-/gender-matched controls for 90 min followed by the measuring the ROS production using a fluorescence plate reader (Ex:Em = 485 nm/535 nm) as described in materials and method section. As we have previously reported [18], a histogram analysis indicated that the ROS producing capacities of the CFS + OI serum samples were significantly higher than the control serum samples (Figure 5A) [t_1,18_ = 2.541; *p* < 0.05 (=0.0205)]. Since all data points in both groups were normally distributed (Figure 5B), we performed a parametric test to study the significance of mean between groups. Interestingly, a Pearson correlation statistical analysis in this cohort revealed that a moderately positive correlation existed between BH4 levels and ROS production, suggesting that elevated BH4 levels in CFS + OI serum samples might be associated with an increased oxidative response (Figure 5C).

Taken together, our current manuscript highlights that CFS patients are associated with elevated levels of BH4 in serum and that upregulation was more significantly observed in CFS patients with OI. Moreover, the elevated BH4 might be associated with the oxidative stress response highlighting a potential role of BH4 metabolism in the pathogenesis of CFS and CFS with OI (Figure 5D).

## 3. Discussion

Tetrahydrobiopterin (BH4) biosynthesis is a tightly regulated metabolic process in health, and its abnormal expression is linked to a series of diseases. Its deficiency leads to the metabolic deficit of amino acid metabolism [19,20], the impairment of dopamine regulation [21], and the impairment of blood flow [22,23] and is associated with hypertension, diabetes, atherosclerosis, aging, and other metabolic disorders. On the other hand, upregulated BH4 biosynthesis impairs mitochondrial energy production [24], induces oxidative stress [25], and augments autophagy impairment [26]. Elevated BH4 is also linked to a reduction in blood pressure [27] by inducing the vasodilative response via the activation of endothelial NOS [28]. Therefore, upregulated biosynthesis of BH4 might directly promote the reduction in blood pressure or hypotension. In fact, a study suggests that the deficiency of BH4 attenuates LPS-induced hypotension in mice [23]. Until now, the role of BH4 metabolism has not been explored in CFS patients, in particular CFS patients with OI. Therefore, our goal was to assess the serum BH4 expression in this patient population. Surprisingly, our ELISA analysis revealed that serum samples from CFS patients had significantly higher levels of BH4 compared to age- and gender-matched healthy controls control subjects. Accordingly, Spearman correlation statistics indicated that there was no significant correlation of BH4 upregulation with respect to either age and gender, nullifying the possibilities of either age- or gender-related impairment of BH4 biosynthesis in these patients.

Next, we observed that the serum samples of CFS + OI patients also displayed a strong elevation of BH4 compared to the serum samples of age- and gender-matched controls. Our cumulative column statistics revealed that 7 out of 10 CFS + OI patients had remarkably high levels of BH4, suggesting that the serum elevation of BH4 might be linked to the pathogenesis of OI. The result was further corroborated when a subset of CFS + OI patients with small fiber polyneuropathy (SFN) also displayed high expression of BH4 in their serum samples compared to the age- and gender-matched controls. Interestingly, a comparison of serum BH4 levels between CFS only and CFS patients with OI demonstrated that there was significantly elevated levels of BH4 in OI patients.

How does elevated BH4 contribute to the pathogenesis of ME/CFS? There are multiple pathways by which elevated BH4 could directly contribute to the pathogenesis of ME/CFS (Figure 5D). First, BH4 upregulation is reported to cause mitochondrial impairment of energy metabolism via inhibition of complex I and IV of the electron transport chain, which results in the reduction in mitochondrial membrane potential, the release of cytochrome C and apoptosis [24]. BH4 exposure is also reported to cause oxidative stress [29]. Although physiologically BH4 supports endothelial NO production by eNOS activation, oxidation of BH4 by superoxide uncouples it from eNOS, which renders oxidative stress to cardiac tissue and affects cardiovascular health (Figure 5C). In support, our results demonstrated that the elevated levels of BH4 in CFS + OI serum samples had a positive correlation with ROS-inducing abilities. Moreover, BH4 also activates mammalian targets of rapamycin complex1 (mTORC1) kinase to impair autophagy. Previously, we reported that the activation of mTORC1 and subsequent impairment of autophagy might be associated with ME/CFS pathogenesis. Both mTORC1-dependent autophagy impairment [18] and mitochondrial deficiency of energy metabolism [30] have been reported to contribute to the pathogenesis of ME/CFS. Therefore, an elevation of the BH4 level in serum samples of ME/CFS patients could induce oxidative stress and augment eNOS uncoupling, followed by cardiovascular stress, hypotension, mitochondrial toxicity, and autophagy impairment in CFS patients.

## 4. Materials and Methods

*Sample acquisition:* Sample acquisition, deidentification, and storage plan were discussed before [18]. Briefly, Blood samples and questionnaire data were collected under the supervision of Dr. Daniel Peterson (Sierra Internal Medicine, Incline Village, NV, USA) (Western IRB) #20201812. Blood samples were centrifuged, and serum samples were aliquoted and then immediately frozen at −80 °C. Each sample was given a unique identification number and recorded both in a notebook and Microsoft Excel with date and signature per the IRB-approved protocol. Samples were then delivered to our research facility in Wisconsin on dry ice overnight. Upon receipt, samples were processed and assayed immediately. Questionnaire and de-identified clinical data are stored in a secure, limited access Redcap sever database managed by the Research Staff and Clinical Fellow at Sierra Internal Medicine. Patient records were maintained in accordance with privacy policy guidelines set by Sierra Internal Medicine. Subsets of ME/CFS patients in the OI and the OI + SFN groups were established using blind chart review, and post laboratory analysis were confirmed by the clinical PI. The four cancer patients were blindly included in this analysis and served as a disease positive control.

The diagnostic criteria for OI included pertinent findings (cardiac tilt table results) from clinical chart review. In addition, at the time of blood collection, the clinical fellow completed a NASA lean analysis. SFN diagnosis was established via a clinical chart review and histological examination of a skin punch biopsy collected from the pre-tibial region. Skin biopsies were independently assessed at the Massachusetts General Hospital for the morphometric quantitation of epidermal nerve endings to study the neurite density of skin surface area.

*BH4 Competitive ELISA*: BH4 level was measured and quantified in serum samples of ME-CFS patients and age-matched healthy controls per the manufacturer’s instructions (Human BH4 competitive ELISA kit; Vendor: MyBioSource, Inc, San Diego, CA 92195, USA; Cat # MBS733839). The kit was validated in our lab (Perkin Elmer Victor X3 multimodal plate-reader, Perkin Elmer, Waltham, MA, USA) and had a standard curve (R^2^ = 0.98) and sensitivity as low as 1 ng/mL concentration with respect to the positive control. Briefly, serum samples were two-times diluted with assay diluent (or 1× PBS) and then pipetted on a microplate that was pre-coated with polyclonal anti-BH4 antibody. After 30 min of incubation with the sample, BH4-HRP conjugate was added to the plate and incubated for an additional 1 h at 37 °C. The residual serum samples were decanted and washed on the plate with 1× wash buffer (alternatively, with 1× PBS-T) three times for five minutes each using a multi-channel pipette. After washing, each well was reacted with TMB substrate for 10 min in the dark. The development of color takes place after 10 min of substrate incubation under shaking conditions. The BH4-HRP conjugate would compete with serum BH4 to bind to the coated-primary antibody. Therefore, the more serum BH4 in a sample, the less BH4-HRP binding would occur and thus generate less color. A total of 66 serum samples (34 control + 32 ME-CFS patients) were acquired (details in “acquisition of samples” section) with proper institutional regulatory guidelines and analyzed under GLP conditions. The sample distribution was random, and the assay was double-blinded, as described under the “sample size justification” section.

*Sample size justification and statistical analysis:* Sample size was justified using the following equation. $Samplesize=n=\frac{z^{2}p\left(1-p\right)}{\varepsilon^{2}}=\frac{1.28^{2}*0.95*\left(1-0.95\right)}{0.05^{2}}=32$. Z is the z score, which is 1.28 for power 0.8. *p* is the population proportion; for a 95% confidence interval, *p* will be 0.95. ε is the margin of error, which is 0.05. Therefore, in order to achieve a significance in the difference of mean between groups at *p* < 0.05, a total of *n* = 32 samples per group were included. For the comparison between control and CFS + OI patients, *n* = 10 samples per group were included. A similar sample size was included while comparing control and control +OI + SFN groups. The sample size calculation was performed based on ε = 0.05 and *p* = 0.99. A non-parametric Mann–Whitney U test was adopted to test the significance of the mean between the two groups whenever the two groups displayed identical and normal data distributions; otherwise, a parametric unpaired *t*-test was performed. For paired analysis between groups, a parametric paired *t*-test was performed if datapoints from both groups were normally distributed; otherwise, a Wilcoxon rank test was performed to evaluate the significance between groups. A Two-way ANOVA with post hoc Bonferroni or Tukey was applied to test the significance of the mean in more than two groups considering sex (male or female) and health (healthy or CFS) as effectors. All statistical analyses were performed in GraphPad Prism version 9.5.1 (733) software (Boston, MA 02110, USA).

*ROS assay:* HMC3 human embryonic microglial cells were grown and passaged with complete DMEM media. Briefly, HMC3 cells were plated in a 96 well-plate for the ROS fluorometric study. Cells were grown to 75–80% confluency and incubated with a DCFDA probe (Cat# ab113851; Abcam, Cambridge, UK) for 45 min at 37 °C in the dark. The standard media was replaced with 100 µL (1:2 dilution = 50 µL serum + 50 µL media) of serum-supplemented media for 90 min to achieve the optimum fluorescence value. At the end of the incubation period, media was aspirated and then measured at the excitation emission using the 485 nm and 535 nm filter sets in PerkinElmer Victor X3 plate reader (Perkin Elmer, Waltham, MA, USA).

## Figures and Tables

**Figure 1 ijms-24-08713-f001:**
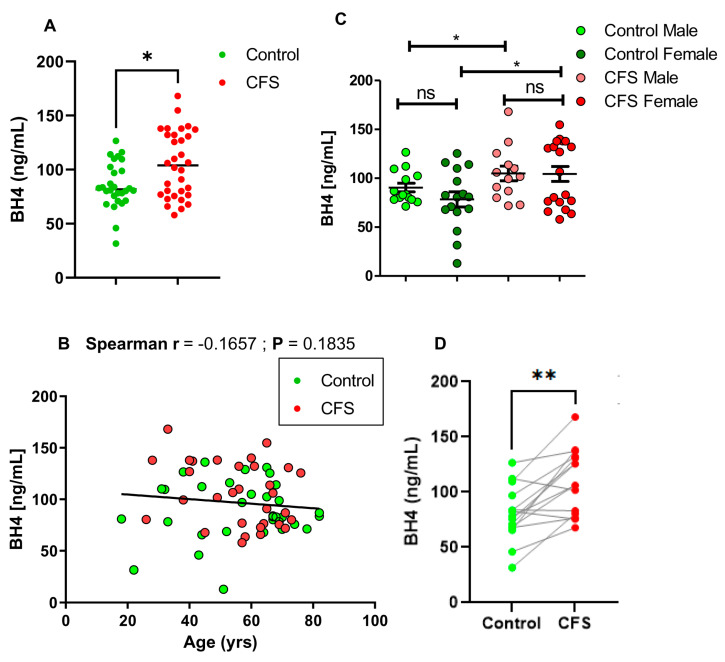
Quantitative analysis of BH4 in serum samples of control and CFS subjects. (**A**) Serum samples were 1:2 diluted by assay diluents and then assayed for BH4 by competitive ELISA method. *n* = 30 control and *n* = 32 CFS subjects were included. The significance of mean was tested by a Mann–Whitney U test between two groups at * *p* < 0.05. (**B**) A non-parametric Spearman correlation analysis was performed to determine the relationship between age and serum BH4 levels of *n* = 62 subjects (Red = CFS and green = control). Trendline was shown through the data points. (**C**) A scatter dot plot analysis compares BH4 levels in control males (*n* = 16; light green), control females (*n* = 18; deep green), CFS males (*n* = 13; light red), and CFS females (*n* = 19; deep red). A two-way ANOVA analysis (effectors are gender and disease) showed that irrespective of gender difference, the BH4 level is always higher in CFS compared to control. To compare the significance of mean between groups, a normality distribution test was performed, which indicates the first three groups were normally distributed. However, CFS female group failed the normality test. As a result, both parametric (unpaired *t*-test) and non-parametric tests (Mann-Whiney U test) were performed. No significance was observed neither between control males and control females [unpaired *t*-test; t_1,32_ = 1.867; *p* > 0.05 (=0.07)], nor between CFS males and CFS females [MWU test; *p* > 0.05 (=0.9654); U = 123]. However, both CFS males and CFS females had significantly higher levels of BH4 compared to control males (unpaired *t*-test; t_1,27_ = 1.008; * *p* < 0.05; =0.0384) and females (MWU test; * *p* < 0.05; =0.0494), respectively. Ns = no significance. (**D**) A pair-wise comparison of BH4 between age- and gender-matched subjects (*n* = 15 pairs). The age and gender of each pair (Control/CFS) were as follows: 82M/76M, 70F/72F, 69M/67M, 64F/64F, 67F/67F, 52F/54F, 33M/32M, 74M/76M, 38M/41M, 22F/26F, 43F/45F, 44M/49M, 70F/69F, 57 F/56F, and 44F/49F. M = male and F = female. Significance was tested by Wilcoxon matched-pairs signed rank test ** *p* < 0.01 (=0.0015) versus control.

**Figure 2 ijms-24-08713-f002:**
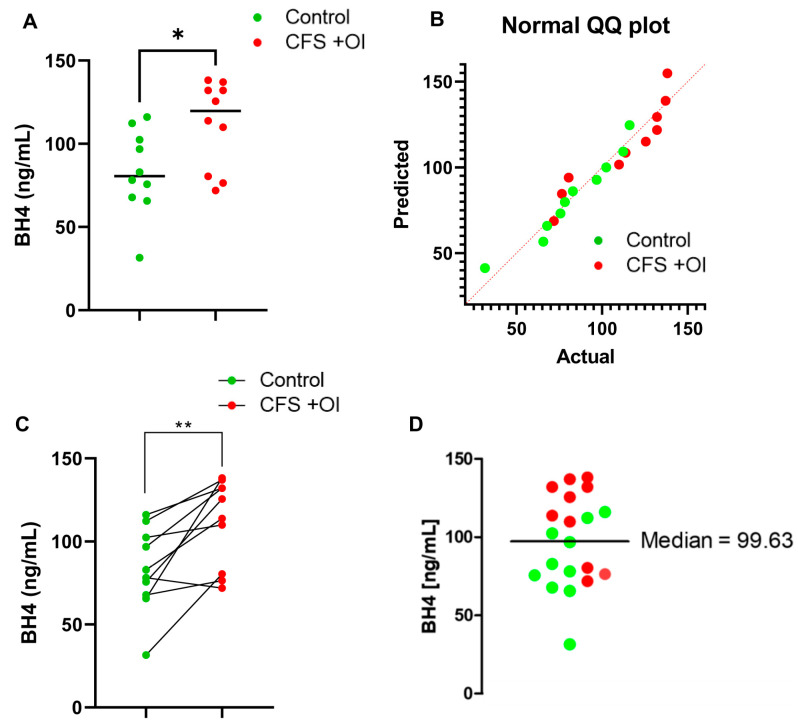
Comparison of BH4 levels between control and CFS + OI subjects. (**A**) A competitive BH4 ELISA assay in 1:2 diluted serum samples of *n* = 10 CFS + OI (red) and *n* = 10 age-/gender-matched control (*n* = 10) subjects. The significance of mean was tested by unpaired *t*-test between the two groups at * *p* < 0.05 (=0.0223). (**B**) A normality distribution of datapoints were analyzed by a Q-Q (Quantile-quantile) plot that showed all datapoints of both groups were normally distributed. (**C**) A scatter plot to evaluate pair-wise comparison of datasets (as summarized in Table 2) between control and CFS + OI subjects (** *p* < 0.01; = 0.004). (**D**) A scatter dot plot analysis was performed to test how many subjects had higher BH4 levels above the median BH4 concentration (99.63 ng/mL as shown by a solid black line). Green dots are controls (*n* = 10), and red dots are CFS + OI subjects (*n* = 10). OI = Orthostatic Intolerance. Results are the mean of three different experiments.

**Figure 3 ijms-24-08713-f003:**
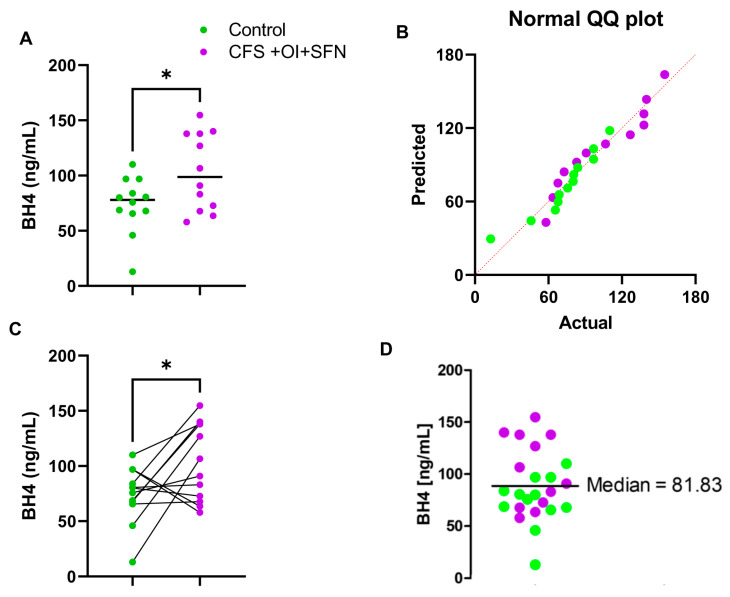
Assessment of difference in BH4 levels between control and CFS + OI + SFN subjects. (**A**) A competitive BH4 ELISA assay in 1:2 diluted serum samples of *n* = 12 CFS + OI + SFN (purple) and *n* = 12 age-/gender-matched control (*n* = 12) subjects. The significance of mean was tested by unpaired *t*-test between the two groups at * *p* < 0.05 (=0.0269). (**B**) Q-Q plot revealed normal distribution of all data points. (**C**) A gender- and age-based paired analysis of BH4 between control and CFS subjects. Paired *t*-test (* *p* < 0.05) confirmed that the CFS serum samples had significantly elevated BH4 compared with the control serum samples. (**D**) A scatter dot plot analysis was performed to test how many subjects had higher BH4 levels above the median BH4 concentration (81.83 ng/mL as indicated by a solid black line). Green dots are controls (*n* = 10), and purple dots are CFS + OI+SFN subjects (*n* = 12). SFN = small fiber polyneuropathy. Results are the mean of three different experiments.

**Figure 4 ijms-24-08713-f004:**
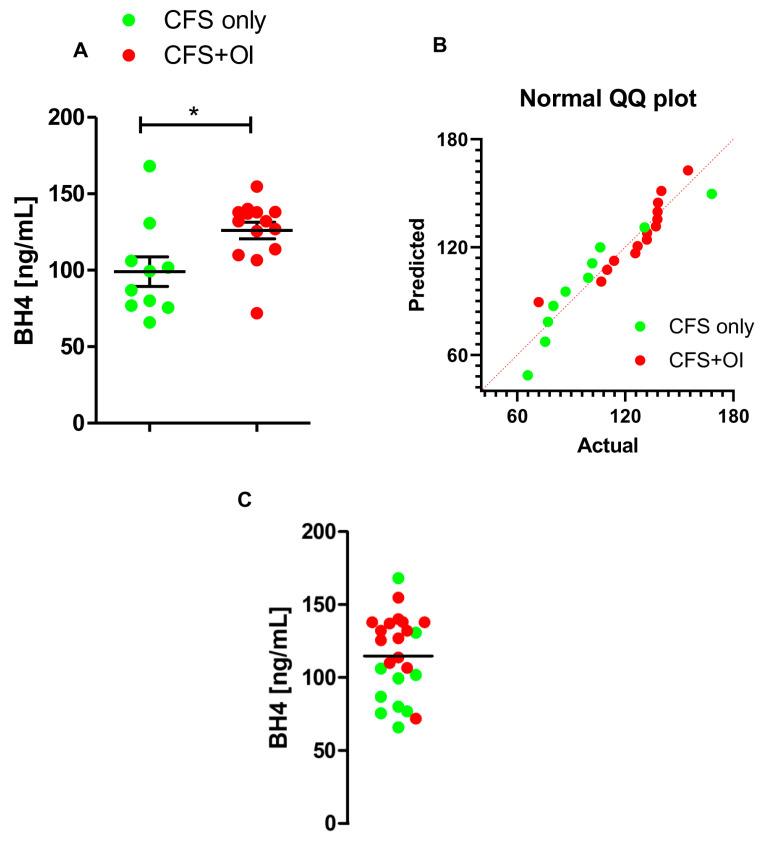
Comparison of BH4 levels between CFS only and CFS + OI subjects. (**A**) Serum samples were 1:2 diluted by assay diluents and then assayed for BH4 by competitive ELISA method. *n* = 10 CFS only control and *n* = 14 CFS + OI subjects were included. The significance of mean was tested by unpaired *t*-test between the two groups at * *p* < 0.05 (=0.0166). (**B**) Q-Q plot represents the normal distribution of data points from both the groups. (**C**) A scatter dot plot analysis was performed to test how many subjects had higher BH4 levels above the median BH4 concentration (119.7 ng/mL as shown by a solid black line). Green dots are CFS only controls (*n* = 10), and red dots are CFS + OI subjects (*n* = 14). OI = Orthostatic Intolerance. Results are the mean of three different experiments.

**Figure 5 ijms-24-08713-f005:**
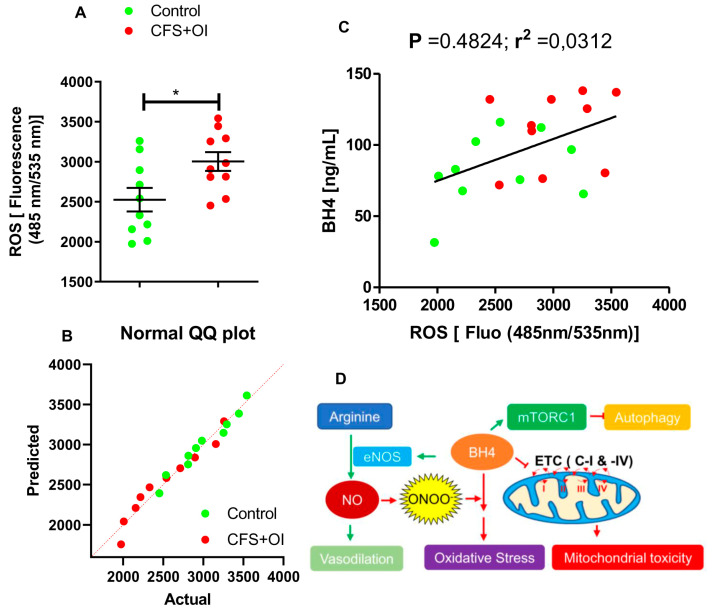
Correlation between BH4 levels and ROS-inducing abilities in serum samples of CFS + OI subjects. (**A**) Control serum and CFS + OI serum-supplemented media was applied on DCFDA-transfected HMC3 human microglial cells for 90 min and then assayed for ROS using the fluorometric method [Ex:Em = 485 nm/535 nm]. *n* = 10 Control and *n* = 10 CFS + OI subjects were included. The significance of mean was tested by unpaired *t*-test between the two groups at * *p* < 0.05 (=0.0205). (**B**) Q-Q plot was derived to display normal distribution of data points. (**C**) A Pearson correlation analysis was performed to determine the relationship between ROS-inducing ability (as measured at fluorescence of 484 nm/535 nm) and serum BH4 levels of *n* = 20 subjects (Red = CFS + OI and green = control). Trendline was shown through the data points. (**D**) The potential role of BH4 in CFS was summarized in a sketch. BH4 induces the production of nitric oxide (NO) via catalytic activation of endothelial nitric oxide synthase (eNOS) enzyme. Endothelial NO stimulates vasodilation and facilitates hypotension. BH4 is prone to be oxidized in the presence of reactive nitrogen species (ONOO-) and induces oxidative stress. BH4 also inhibits electron transport chain (ETC) in complex-I and -IV (C-I and C-IV) steps and triggers mitochondrial toxicity. Another possibility is the activation of mTORC1 kinase complex and inhibition of autophagy.

**Table 1 ijms-24-08713-t001:** Age, gender, ethnicity, and disease conditions of included subjects in the study.

Control (*n* = 34)					CFS (*n* = 32)				
Study ID	Age	Gender	Ethnicity	Condition	Study ID	Age	Gender	Ethnicity	Condition
329502	82	M	white	Healthy	211103	67	M	white	CFS
339105	70	F	white	Healthy	381104	38	M	white	CFS
939209	69	M	white	Healthy	201105	72	F	white	CFS
699106	68	F	white	Healthy	921106	69	F	white	CFS
229403	68	F	white	Healthy	991108	71	M	white	CFS
809208	67	M	white	Healthy	191109	57	F	white	CFS
499205	67	F	white	Healthy	701110	63	F	white	CFS
169316	65	M	white	cancer	661111	73	M	white	CFS
309110	64	F	white	Healthy	341112	33	M	white	CFS
829306	60	F	white	cancer	861113	49	M	white	CFS
909302	58	F	white	cancer	691114	49	F	white	CFS + OI
739109	57	F	white	Healthy	541201	41	M	white	CFS + OI
299114	53	F	white	Healthy	391203	61	F	white	CFS + OI
149304	52	F	white	Healthy	771204	56	M	white	CFS + OI
759113	45	M	white	cancer	951205	64	F	white	CFS + OI
109201	44	M	white	Healthy	641206	76	M	white	CFS + OI
639305	44	F	white	Healthy	431207	56	F	white	CFS + OI
219317	43	F	white	Healthy	331209	71	M	white	CFS + OI
129104	33	M	white	Healthy	281210	66	M	white	CFS + OI
369112	32	M	white	Healthy	661215	26	F	white	CFS + OI
129312	31	F	white	Healthy	261302	58	F	white	CFS + OI + SFN
169504	82	M	white	Healthy	791304	54	F	white	CFS + OI + SFN
899108	69	M	white	Healthy	811305	45	F	white	CFS + OI + SFN
669206	78	M	white	Healthy	321306	57	F	white	CFS + OI + SFN
419111	74	M	white	Healthy	971309	60	F	white	CFS + OI + SFN
249311	70	F	white	Healthy	851310	65	F	white	CFS + OI + SFN
889210	65	M	white	Healthy	221311	67	F	white	CFS + OI + SFN
559310	66	F	white	Healthy	751312	28	F	white	CFS + OI + SFN
559107	68	M	white	Healthy	901313	65	M	white	CFS + OI + SFN
539103	69	M	white	Healthy	181316	63	M	white	CFS + OI + SFN
789207	51	F	white	Healthy	841317	40	F	white	CFS + OI + SFN
609405	38	M	white	Healthy	1004	40	F	white	CFS + OI + SFN
579315	18	F	white	Healthy					
431315	22	F	white	Healthy					

CFS = chronic fatigue syndrome; OI = Orthostatic intolerance; SFN = small fiber polyneuropathy.

**Table 2 ijms-24-08713-t002:** Comparison of BH4 levels between controls and age- plus gender-matched CFS + OI patients. Results are the mean ± SEM of three independent assays.

Control			CFS + OI		
Age	Gender	BH4 (ng/mL)	Age	Gender	BH4 (ng/mL)
44	F	65.63 ± 4.098	49	F	138.152 ± 24.332
44	M	112.296 ± 12.113	41	M	137.053 ± 13.165
57	F	96.841 ± 16.528	61	F	132.01 ± 18.109
65	M	102.42 ± 9.287	56	M	109.89 ± 11.98
64	F	67.852 ± 11.121	64	F	76.494 ± 23.4
74	M	75.669 ± 14.225	76	M	125.53 ± 4.164
53	F	116.068 ± 8.119	56	F	132.099± 8.119
69	M	78.222 ± 4.6	71	M	71.916 ± 12.01
68	M	82.914 ± 23.161	66	M	113.778 ± 4.33
22	F	31.556 ± 7.110	26	F	80.444 ± 16.12

**Table 3 ijms-24-08713-t003:** Comparison of BH4 levels between controls and age- plus gender-matched CFS + OI + SFN patients. Results are the mean of three independent assays.

Control			CFS + OI + SFN		
Age	Gender	BH4 (ng/mL)	Age	Gender	BH4 (ng/mL)
57	F	96.84128	58	F	63.654
51	F	12.85	54	F	106.617
44	F	65.63	45	F	67.66872758
57	F	96.84	57	F	57.975
64	F	67.851	60	F	140.0630005
67	F	83.9012	65	F	154.8251948
68	F	80.614	67	F	83.041
31	F	110.074	28	F	137.8619147
74	M	75.66941	65	M	90.87
67	M	80.166	63	M	72.790
43	F	45.876	40	F	126.8371241
52	F	68.78	40	F	137.98

## Data Availability

There is no electronic datasheet associated with this paper. No data in electronic repository.

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
