# Peer review of "Detection of Elevated Level of Tetrahydrobiopterin in Serum Samples of ME/CFS Patients with Orthostatic Intolerance: A Pilot Study"

_ijms, 2023, doi:10.3390/ijms24108713_

Round 1

Reviewer 1 Report

The writing is great. Below are my comments

Introduction

While the introduction is good, please use the final paragraph writing your objective instead of what you did.

Method:

Why didn't you match the subjects? Since you collected data from a physician's practice, you could easily age match the participants. 

Why did you not run Mann-Whitney U tests to identify differences between groups for age? 

Based on the distribution of the data, why did you run parametric tests? 

When did you run ANOVAs? Was it for sex differences/ I can't tell based on your write-up. You should have run a Kruskal-Wallis, since the data is not normally distributed (based on the figures you have provided).

Although I don't expect your results to change much, the statistical analyses are incorrect and need to be re-done. 

I cannot evaluate the discussion until the analyses are conducted correctly.

Author Response

Response to Reviewer#1:

After careful assessments of comments made by Reviewer#1, we performed all statistical analyses, tested the normality distribution of data points from all groups, and incorporated parametric and nonparametric tests to calculate the significance between groups. Please find our response to reviewer #1 as follows.

Introduction

The objective was added in the last but one paragraph and highlighted with the track.

Method

Based on reviewer#1’s recommendation, we have incorporated the non-parametric test whenever possible. For Fig. 1A, we first performed a normality test. The CFS group did not pass the normality test. Accordingly, the analysis was re-performed, the t-test was replaced with a nonparametric Mann-Whitney U test, and necessary adjustments were incorporated in the results, figure, and figure legend. The changes were displayed by track.

Similarly, for Figure B, Pearson correlation analysis was replaced with Spearman correlation analysis. However, the outcome remained the same as no significant correlation was observed between BH4 and age. The necessary changes were made by track in results, figures, and figure legends.

The justification of statistical analysis for Figure 1C is as follows.  

In the current analysis,  the group number is more than two (=4) with two independent variables or effectors ( gender and health condition) regulating the expression of BH4. Therefore, we did not do the MWU test. Instead, we ran the 2-way ANOVA tests.  Moreover, the first three groups were normally distributed, and the last group was not. Therefore, the significance assessment between the first three groups will be both parametric (control male vs control female; control male vs CFS male) and nonparametric ( control female vs. CFS female and CFS male vs CFS female). The relevant discussion was made, and changes were incorporated in the result section and figure legend 1.

Since the cohort is tightly regulated in terms of gender, age, and geographical area, we are intrigued by the idea of nonparametric pair-wise comparison analysis. In fig. 1D, we have incorporated a new analysis comparing BH4 levels between age- and gender-matched subjects. The significance was analyzed by the Wilcoxon matched-pairs signed rank test. The necessary changes were made in the results, figures, and figure legends.

Fig. 2A is the analysis of BH4 between control and control+OI groups. Both groups have equal sample sizes and based on the normality test, both data sets are normally distributed (P>0.05). Therefore, the parametric test is the right choice. The Q-Q normality distribution plot was incorporated to display the distribution of data (Fig. 2B) . Similarly, we also performed a pair-wise comparison of BH4 between subjects with similar ages and gender (Fig. 2C) The necessary changes were discussed in the results, figure, and figure legends.

Similar normality distribution analyses were performed in Figures 3, 4, and 5. The significance of the mean between groups was accordingly evaluated. All changes were incorporated in the results, figure, and legend section ( please see track).

Reviewer 2 Report

The authors submitted a research article in which they reported that the chronic fatigue syndrome patients had elevated levels of BH4 in serum that it might be associated with the oxidative stress response. The aim of the study is clear. The manuscript has a logical structure and contains well-written subsections. The tables and figures are clear and legible. The conclusive part seems to be informative. Although the results of the study are impressive, I would like to make several comments to discuss.

1. Patients study population. This subsection of the Results should be more thoroughly described that it is. Please, re-write this subsection so that the protocol of the study would clearly report and basic characteristics of the patients would compare. The flow chart is welcome.

2. The authors should give more information of the clinical staus of the patients along with comorbidities and medications, which can intervine in the study results.

3. The authors used descriptive statistics and Pearson correlation, whereas the data might be compared with MANCOVA with comorbidity factors as variables

Author Response

Response to Reviewer#2:

After careful assessment of comments made by reviewer#2, we reformatted the paper based on more details about patients and their diagnostic criteria. We have also performed detailed statistics throughout the manuscript. The following responses were included to satisfy reviewer #2.

  1. Patient study population. The details were added in paragraph #2 under the “subjects included in the study” subsection of Results.
  2. Necessary info regarding the clinical status of these patients was mentioned in paragraph#3 under the “subjects included in the study” subsection of Results.
  3. The statistical analyses of the entire paper were repeated, modified, and changes were shown in the track option in the results section, figures, and figure legends.

Round 2

Reviewer 1 Report

I appreciate the authors addressing my concerns. I also appreciate the authors adding their objectives in the introduction.

There are a few minor errors in punctuation (i.e. missing commas) and grammar.